# Perceived stress and well-being of Polish migrants in the UK after Brexit vote

**Klaudia Martynowska**[1]☯, **Tomasz Korulczyk**[2]☯, **Piotr Janusz Mamcarz**[3]☯*

**1** Department of Special Education, The John Paul II Catholic University of Lublin, Lublin, Poland,
**2** Department of Organizational and Management Psychology, The John Paul II Catholic University of Lublin, Lublin, Poland, **3** Department of Emotion and Motivation Psychology, The John Paul II Catholic University of Lublin, Lublin, Poland

☯ These authors contributed equally to this work.
* pmamcarz@kul.pl

**Data Availability Statement:** SPSS and M+ datasets and syntaxes related with the study are published and available here: https://doi.org/10.6084/m9.figshare.12357593.v1.

## Abstract

### Objective

This study aimed to investigate factors affecting personal well-being of Polish immigrants living in the UK in the face of a significant political change—the Brexit vote. We measured perceived changes in attitude or behaviour of supervisors and co-workers, respondents' perceived stress, and its outcomes such as psychological well-being and intention to leave the UK after the Brexit vote.

### Method

551 Polish migrants residing in various regions of the UK took part in the study in the form of Qualtrics online survey. We used self-report measures: Perceived Stress Scale, The Satisfaction with Life Scale, Scale of Psychological Well-being.

### Results

The most of the respondents did not notice any change in the attitude or behaviour of the supervisor (81%) or co-workers (84%), and only a small percentage of the participants reported negative changes in attitude or behaviour of supervisors (9%) and co-workers (14%). Also, negative change in attitude or behaviour of supervisors or co-workers are associated with perceived stress, which inturn is linked with intention to leave the UK, psychological well-being and life satisfaction.

### Conclusion

Polish and British co-existence in a workplace setting has not changed much after the Brexit vote.

**Funding:** The author(s) received no specific funding for this work.

**Competing interests:** The authors have declared that no competing interests exist.

## Introduction

Brexit has not only signified a political change but, more than anything, a major social change in immigrants' life. It is reflected in the statistics which show that there has been a rise in the number of European citizens leaving the UK since the Brexit vote. In 2016, a 84.000 drop in migration to the UK—the lowest level for nearly three years—was driven by a 40.000 rise in emigration compared to 2015. According to The Guardian—a leading British newspaper—the immigration drop was partly caused by a dropping number of 25.000 Poles and other Eastern Europeans—citizens coming to work in Britain, probably affected negatively by the referendum vote [1], and a 16.000 rise in the number of them leaving the UK. Generally speaking, the Brexit vote, among others, has elucidated the right-wing anti-globalisation movement which may result in negative social behaviours such as xenophobia reflected in the experience of minorities [2]. The anti-immigrant rhetoric of some political leaders in the UK has led to a rising number of racially aggravated offences towards immigrants after Brexit vote [3, 4]. The evident examples included cards saying "Leave the EU/No more Polish vermin" left outside primary schools in Huntingdon, Cambridgeshire [5], and the tragic example of the murder of a middle-aged Polish factory worker—Arkadiusz Jóźwik who after being heard speaking Polish was punched to the ground [6]. Racism, racially motivated hostility, and xenophobia in the UK are not new phenomena. Hate crimes and racially motivated hostility strongly correspond to shifts in demographics, political, and economic trends [7] Brexit vote. However, the Brexit results "legitimised" these and caused a significant increase in their occurrence [8] towards not only Polish but also other minorities living in the UK [9, 10]. Despite a growing interest in the after Brexit vote consequences with many papers exploring racist hate crimes as a posttraumatic stress disorder and victimisation [1], there is not much evidence presenting the psychological effects of this political decision experienced by migrant employees. Also, while there is a regular survey to study the psychological well-being (PWB) of UK citizens, there is a considerable lack of attention paid to the PWB of the immigrants, also in the scientific terms [4]. In an organisational context, racially motivated hostility may take more latent or open form. The first one may be expressed in the form of a change of attitude into more hostile and can be observed through indirect unfriendly behaviours. The latter one is manifested through incidents of discrimination, hateful behaviours and may even take a form of workplace harassment. Therefore, the present study aims to examine a possible change in attitude or behaviour of supervisors (NCS) or co-workers (NCC), perceived stress (PSS) and its consequences on well-being of migrant employees.

### Perception of Poles living in the UK over time

The EU Succession of Poland and seven other countries (A8) on 1st of May 2004 triggered a mass economic migration within the EU. While the majority of existing EU members restricted access to their labour markets for citizens of newly joined countries, the UK as one out of three countries allowed legal work immediately after EU enlargement. This political decision was made due to significant shortages in the labour market in low-paid and low-skilled jobs [11]. Many people experiencing high unemployment, lack of career opportunities, and low wages in Poland decided to leave for the UK to work and settle down [12]. Within less than a decade, the Polish-born population enlarged almost ten times, reaching a number over half of a million. The statistics show that 15 years after the EU succession, there are over a million Polish-speaking people in the UK [13]. It constitutes about 2% of the whole UK population, which makes Polish people the largest non-British nationality residing in the UK. At the time of EU succession, Polish people were considered as hardworking, honest and acknowledged by British people for their contribution in the recent history of England, in other words,

they were perceived as desirable employees on the UK labour market. Yet, the image of Polish people has gradually changed after the UK's market suffered from economic problems due to the crisis in 2008. At the very same time, the Polish population enlarged from 50.000 to about a million. Being an EU member, UK was bound by freedom of movement law which at some point was regarded as the main political reason causing problems on the UK's market. To get a vote of approval, leaders of some political parties claimed free migration as a challenging issue while intensifying their anti-immigrant rhetoric. In 2016, the Leave campaign argued that leaving the EU would allow Britain to "take back control of its borders". The other politician—Amber Rudd during the Conservative Party's conference in October 2017 said that foreign workers should not be "taking jobs that British people could do" echoing Gordon Brown's "British jobs for British workers" remark in 2007 [8]. Not only politicians contributed to the change of perception of Polish people living in the UK, but it was media who played a leading role in this process, mainly three of them: The Sun, the Express, and the Daily Mail [14]. As a result, the European Commission against Racism and Intolerance in 2016 denounced the UK newspapers for using offensive, discriminatory language.

## Social consequences of Brexit vote

After the conservative party won the election in 2015, under the pressure of society and media, the prime minister, David Cameron, announced the Brexit referendum. Surprisingly, most electors, yet slightly over the half, with the majority of votes of the older population decided to leave the EU. This situation sparked a large number of hateful incidents towards minority groups living in the UK. The analysis of over a hundred of racist violence incidents after Brexit vote supported the link between the rhetoric of politicians and behaviour of perpetrators [9]. By engaging in hostile behaviours, the offenders may feel like they are "executing" the results of the vote. They perceive their behaviour as totally justified and believe their acts of aggression are even patriotic. Comparing the situation of Polish people in the UK after Brexit vote and the minorities in the US after 9/11, one can observe apparent similarities. Craig-Henderson and Sloan expressed it as follows: "Since the September 11 terrorists attacks, the Civil Rights Division of the Justice Department has noted a dramatic increase in the number of violent hate crimes directed at people who are perceived to be Arab, Muslim, or simply Middle Eastern. In general, Middle Easterners have been targeted because they are perceived to share membership in the ethnic group to which the perpetrators of the terrorist attacks belong. There is evidence that the Americans who have perpetrated anti-Middle Eastern hate crimes in the wake of 09/11 and have been apprehended see themselves as strongly patriotic and believe their aggression to be retaliatory and justified" [15], p. 482. Polish migrants have been targeted due to their membership to an ethnic group considered a "threat" to the UK citizens. However, this threat has not been physical but rather economic [16]. Within 15 years as a result of strong political discourse, the image of Poles has changed from desirable members of society to the members of the threatening group. It is reflected in both public and organisational environments which may be manifested in a negative change in attitude and hostile behaviour in the workplace. The arising question here is whether the collaborative work of Polish and British people for more than a decade has not built a bond strong enough to overcome the stereotypical perception of Poles as threatening minority group.

## A negative change in attitude or behaviour of supervisors and co-workers

In an organisational setting, employees view their work relationships either with the organisation or its authorities as social exchanges [17, 18]. In the light of uncertainty coming from an unstable political situation after Brexit vote, migrants may project their experiences of societal

change in attitude on their work relationships. The change of attitude may relate to the fairness of judgments. Thau, Aquino, and Wittek [19] have extensively researched this concept with antisocial and hostile behaviours, in the framework of uncertainty management theory (UMT) which specifies the mechanisms of fairness judgments affecting work behaviours [20]. Following UMT, employees focus on fairness information, especially when they face uncertainty. It may relate to the trustworthiness of the authority in a decision-making situation [21]. The relational model of procedural justice, fair procedures have an impact on relational bonds among people and group authorities, e.g., supervisors, managers [22]. Fair procedures reflect status to members of an organisation by communicating to the employees that "if we treat you fairly, we must care about you or respect you." [23]. Finally, there are some possible boundary conditions for the well-established empirical relationship between justice perceptions and antisocial work behaviour [24–26]. On the other hand, uncertainty may be linked with instability, both financial and emotional, as a result of the Brexit vote for the minority groups who mostly migrated to the UK. On this basis, the following hypothesis is set (H1): *Respondents observed the negative change in attitude or behaviour of supervisors and co-workers after Brexit vote*.

## Organisational policy and perceived stress

Significant political changes have socioeconomic consequences which are experienced by organisations and institutions and have a substantial impact on the company's internal policy. Landells and Albrecht [27] have undertaken an attempt to develop a richer understanding of how employees perceive organisational politics in contemporary organisational contexts, in positive and negative terms. The findings, among others, revealed that political behaviours were described by five established bases of organisational power: connection power, information power, coercive power, positional power, and personal power. It leads to the assumption of behaviour displayed by employees on various organisational levels exert a sense of power. Moreover, organisational politics have direct, moderated, and mediated effects on stress-related outcomes experienced by employees [28, 29]. An organisational policy may be proactive or reactive. Reactive behaviours such as avoidance of action, avoidance of responsibility, strategy of passiveness appear more frequently in of the face of change. For instance, an employee perceives a change of rule as a threat that may harm his or her interests and thus acts to mitigate the losses in the future. Every abnormality in the work environment can be a trigger to stress, fear, feeling of threat, and other negative cognitive-emotional states. According to the Process Model of Organizational Politics and Stress [30] which underlines the role of perception of organisational politics, political behaviour, and political skills in determining stress-related outcomes, including psychological and physiological well-being as well as job burnout, the following hypothesis is set (H2): *A negative change in attitude or behaviour of supervisors and co-workers increases the level of perceived stress*.

## The role of social support in stress reduction

In the face of demanding life situations imposing major changes, individual's personal resources are of prime importance as well-being and adjustment capabilities depend on them. Hobfoll [31] indicated 74 different types of resources grouped into internal and external ones. The first group of resources embrace values (e.g., development, life, family), psychological capital (e.g., coping with stress, emotional intelligence, the locus of control, optimism, hardiness, self-esteem) and personality traits. External resources, also understood as environmental, may vary with regard to living conditions of an individual. In a work context, positive environmental resources comprise autonomy, constructive feedback, rewards, social support and a positive change in behaviour, and attitude of supervisors/co-workers. A rich vein of contemporary

scholarship sustains that these personal and environmental factors may reduce stress and enhance the level of PWB [32–34]. On this basis, the following hypothesis is set (H3): *Social support moderates the relationship between the negative change in attitude or behaviour of supervisors and co-workers and perceived stress.*

## Perceived stress and its outcomes

Stress-related variables have been widely researched in occupational and organisational settings [35, 36]. Perceived stress may result in negative consequences as an effect of complex interactions among many variables (e.g. psychological wellbeing, physical health and job satisfaction, [37]). Despite physical, psychological and behavioural outcomes, the research indicates a strong relationship between PSS and intention to leave an organisation [38]. Specifically, PSS affects intention to leave both directly and indirectly [39, 40]. The prevailing definition of intention to leave encompasses an employee's intention to quit a current job position and being excited about looking for another job in the near future [41]. Although the literature widely explores intention to leave in an organisational context, our study strictly focuses on the intention to leave the UK. A vast amount of migration studies have explored perceived stress in the process or as an effect of moving countries. In particular, [42] verified following domains of post-migration stress: perceived discrimination, lack of host country specific competences, material and economic strain, loss of home country, family and home country concerns, social strain, and family conflicts. Another research has elaborated on the consequences of international migration on the example of Polish migrant workers in Scotland [43]. The results advocate stress factors from the social and work-related perspectives. Identified challenges related to communication and unfamiliarity with the new environment and culture which explicates the fact that cross-border migration imposes vulnerability on migrants as they have to deal with a wide range of adjustment demands. On this basis, the following hypothesis is set (H4): *Perceived stress increases the level of intention to leave the UK.* SWB entails an evaluation of life in terms of satisfaction and balance between positive and negative affect while PWB encompasses perception of engagement with existential challenges of life through six components: self-Acceptance, personal growth, purpose in life, positive relations with others environmental mastery and autonomy [44]. Research has shown the connection of PWB with performance in a work setting [45, 46] reflected in successful relations [47]. On the more negative side, there is a wide range of research investigating complex interaction between stress and PWB [48], especially with a mediating role of variables such as the internal locus of control or social problem-solving. Work-related antisocial behaviours and any other racial/ethnic discrimination may inflict chronic stress associated with the risk of developing mental and physical disorders over time. The latter one has been consistently associated with low PWB represented by poor mental health outcomes among racial/ethnic minority group members such as high levels of depression, anxiety, anger, and symptoms of posttraumatic stress [49]. As well-being is related to positive affect and life satisfaction [50], we decided to examine this relation in more depth. Life satisfaction is defined as a 'cognitive evaluation of one's life as a whole' [51], p. 550 but also as a degree to which an individual judge the quality of their life favourably [52]. Thus, it can be stated that life satisfaction represents an evaluative judgment which has been identified as a distinct construct from well-being. Some researchers argued that life satisfaction judgments are distorted by contextual influences such as small fluctuations in the mood of the respondent [53], or temporary events occurring in a person's life. Others pointed out that major life events or changes in life domains, either positive or negative, may indeed initially influence one's level of life satisfaction or overall subjective well-being, but they exert a short-term impact. People soon adapt to their new circumstances, and their level of life

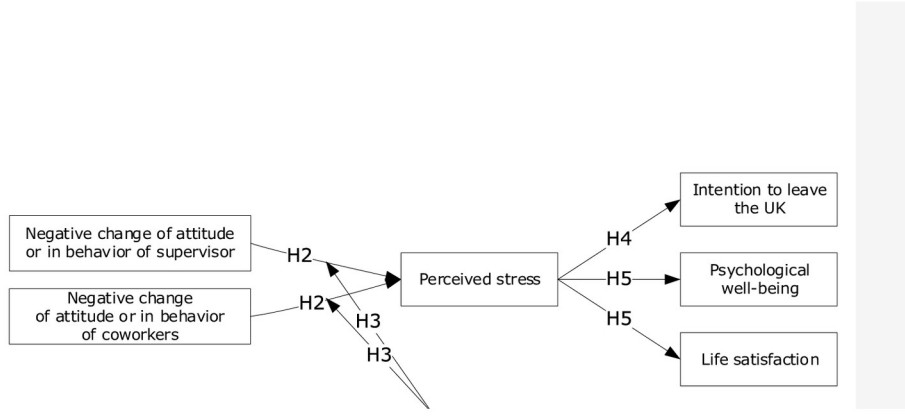

**Fig 1. Postulated model.**

satisfaction returns to the level reported before a significant event or change [54]. In the light of the above-mentioned consideration, we have assumed that information about the UK leaving the European Union may be perceived by immigrants as a major change in their life as some of them would lose the rights to work and live in the UK. It is worth exploring whether this change influenced their life satisfaction at all. On this basis, the following hypothesis is set (H5): *Perceived stress decreases the level of psychological well-being and life satisfaction.* Based on the above-mentioned psychological literature review and presented hypotheses, we postulated the following model (Fig 1).

## Materials and methods

### Sample and procedure

600 individuals were invited to take part in a study, 551 completed the survey (response rate equals to 92%). A majority of the sample comprised women (75%). The average age was 33 years (SD: 7.66), ranging from 17 to 64. Most participants hold a higher education degree (51%) and graduated from high school (41%). The sample's distribution in terms of place of residence is not equal, but it is geographically representative having an approximate distribution to the population of Polish people living in the UK (Fig 2; House of Commons Report 2019). Prior to the research, the approval of the departmental ethics committee (Social Sciences Department) was obtained. The participants rated their financial situation as good (81%), 17% rated a difficult situation which is weekly correlated with PSS (r = .19, p <.05). The average stay in the UK is eight years (SD: 4.63), ranging from six months to 38 years. Data were collected in July 2016 using an online survey platform, Qualtrics, which took up to 10 minutes to complete. The participants were recruited via social media from various Polish community groups on Facebook e.g. "Poles in the UK" [Polacy w UK], "Poles in Northern Ireland" [Polacy w Irlandii Północnej], "Poles in Scotland" [Polacy w Szkocji], "Poles in Wales" [Polacy w Walii], "Ask.us" [Zapytaj.us]. The Facebook post contained an introductory note and a link to the survey. Before the participants completed an anonymous online version of the questionnaires, they were informed of the voluntary nature of their participation, that it was conducted for scientific purposes and it investigated various aspects of human functioning. The participants completed all measures described below and did not receive any reward.

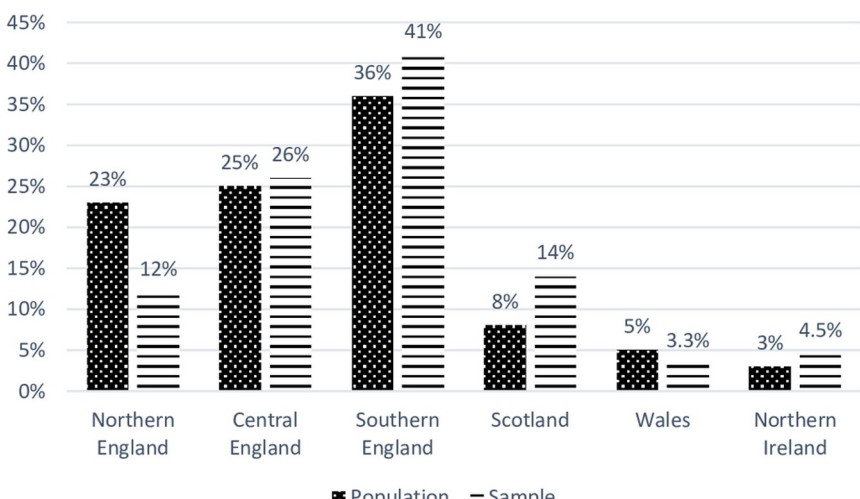

**Fig 2. The distribution of place of residence of Poles living in the UK in the sample and the population.**

## Measures

**The change in attitude or behaviour of supervisors and co-workers** was measured by a generated item "Have you noticed the change in attitude/behaviour of your British supervisors (co-workers, corresponding) towards you after Brexit referendum?" with following single-choice answers from (5) "I have noticed very positive change of attitude or in the behaviour of British supervisors" to (1) "I have noticed very negative change of attitude or in behaviour of British supervisors" (co-workers, corresponding). **Perceived Stress** was measured by Perceived Stress Scale—Mind Garden, which is used for measuring the perception of stress (Cohen, Kamarck, and Mermelstein (1983), in the adaptation of Juczyński and Ogińska-Bulik (2009). The tool measures the degree to which situations in one's life are considered stressful. Items refer to how uncontrollable, unpredictable, and overloading the life is at the present moment. The tool refers to the last month. The items have a general meaning and are relatively free of specific content. The scale consists of 10 items rated on a 5-point Likert-type scale from 0 (never) to 4 (very often). The tool has proven to be valid and reliable ($\alpha$ = .86). **Social Support** was measured by a generated an item: "To what extent do you feel that you are receiving support from family or friends?" with following single-choice answers from 1 (none) to 5 (fully). **Psychological well-being** was measured using the Scale of Psychological Well-being (Ryff 1989; Ryff, Keyes 1995) in the Polish adaptation of Krok (2009). It encompasses six distinct dimensions of wellness (Autonomy, Environmental Mastery, Personal Growth, Positive Relations with Others, Purpose in Life, Self-Acceptance). The scale consists of 42 items rated on a 7-point Likert-type scale from 1 (never) to 7 (always). The alpha coefficient for reliability ranges from.72 to.86. **Life satisfaction** was measured by the Satisfaction with Life Scale (Diener, Emmons, Larsen, and Griffin 1985) contains a 5-items in the adaptation of Jankowski (2015). It is a widely used self-report instrument for evaluating individual global cognitive judgments of one's life satisfaction. It measures a judgmental component of subjective well-being, assesses satisfaction with the respondent's life as a whole, not only specified to the specific domain as health or finances. The tool might be used with a wide range of age groups and applications. Evidence for the validity and reliability $\alpha$ = .81, $r_{test-retest}$ = .86 of the scale is present (Jankowski 2015). **Intention to leave** was measured by a single item: "To what extent did

your intention to return to Poland or move to another country intensify after the Brexit referendum?" with following single-choice answers from 1 (none) to 5 (fully).

## Results

### Preliminary analysis

At first, we tested the data for Common Method Bias [55]. We used Harman's single-factor approach. If a single factor emerges or one general factor will account for most of the covariance among the measures, then it is concluded that a substantial amount of common method variance is present. In our case, we obtained 28 factors or when fixed to a single factor, and it explains 25% of the common variance. Thus, we conclude that common method bias is not present in our study.

### Hypotheses testing

Table 1 reports the intercorrelations, means, standard deviations, and reliabilities of study variables. In general, high correlations among variables like stress, psychological well-being, life satisfaction, social support corresponds with the past reports from the literature.

Hypothesis one stating that respondents observed the NCS and NCC after Brexit vote has been partially supported. Only 14% of respondents noticed NCC, while 81% did not notice any change, and 5% noticed a positive change after Brexit vote. When it comes to the supervisors, only 9% of respondents report, they noticed NCS, 84% did not notice any change, and 7% noticed a positive change after Brexit vote. The following hypotheses (H2 to H5) were tested in the form of introduced model using Structural Equation Modelling (SEM; Fig 3). We used Mplus software to build and estimate the model [56] and employed Maximum Likelihood Robust (MLR) procedure. Hypothesis two stating that the NCS and NCC increase the level of PSS has been supported. The results show a small positive correlation between NSC and PSS ($\lambda$ = .18, p <.001) and NCC and PSS ($\lambda$ = .27, p <.001). Both variables explain 18% of the variance of PSS. To test hypothesis three, we built the second, alternative model including social support as a moderator of the relationship of NCC and PSS ($\chi 2(13)$ = 39.32; $\chi 2/df$ = 3.02; RMSEA = .061[.040—.083], TLI = 898; CFI = 827). The main model fitted data significantly better (Satorra-Bentler Scaled $\chi 2(3)$ = 25.29, p = 001), thus we dropped social support as a moderator of NCC and PSS from the main model (RMSEA = 051; TLI >.87; CFI >.90; see Fig 3). All path coefficients in the main model are significant (p <.001).

**Table 1. Means, standard deviations, and intercorrelations of study variables.**

| No. | Name | M | SD | $\alpha$ | 1 | 2 | 3 | 4 | 5 | 6 | 7 |
|-----|------|---|----|----|---|---|---|---|---|---|---|
| 1 | NCS | 3.01 | .49 | NA | 1 | | | | | | |
| 2 | NCC | 3.07 | .54 | NA | .55*** | 1 | | | | | |
| 3 | Social support 3 | 3.23 | 1.21 | NA | -.11*** | -.12** | 1 | | | | |
| 4 | Perceived Stress | 16.89 | 7.19 | NA | .21** | 21** | -.12 | 1 | | | |
| 5 | Intention to leave the UK | 2.20 | 1.28 | .88 | .21*** | .29*** | -.11* | .35*** | 1 | | |
| 6 | Psychological well-being | 208.57 | 43.62 | .86 | -.19*** | -.23*** | .16** | -.60*** | -.26*** | 1 | |
| 7 | Life satisfaction | 20.43 | 6.30 | .74 | -.08 | -.16 | .15 | -.57*** | -.26** | -.63*** | 1 |

NCS = negative change in attitude or behaviour of supervisors (NCC = co-workers, corresponding). n from 148 to 549.

* p <.05;

** p <.01;

*** p <.001.

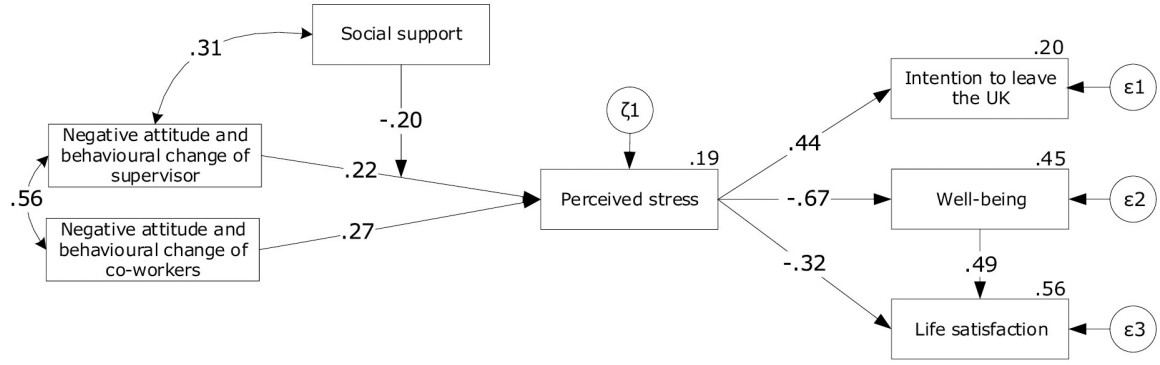

**Fig 3. Structural Equation Model of the effect of after vote NCS and NCC, PSS and its outcomes.**

Hypothesis three stating that the social support moderates the relationship between NCS and NCC and PSS, has been partially supported. The results show a significant, negative correlation between social support and path of NSC and PSS ($\lambda$ = -.13, p <.01). Based on the rejected model, the hypothesis was not supported for the NCC. Hypothesis four stating that the PSS increases the level of intention to leave the UK has been supported. There is a positive correlation between PSS and intention to leave the UK ($\lambda$ = 44, p <.01), Hypothesis five stating that PSS decreases the level of PWB and life satisfaction has been supported. The results show a moderate, negative correlation between PSS and PWB ($\lambda$ = -.67, p <.001), PSS and life satisfaction ($\lambda$ = -.32, p <.001). Hypothesis three stating that the social support moderates the relationship between NCS and NCC and PSS, has been partially supported. The results show a significant, negative correlation between social support and path of NSC and PSS ($\lambda$ = -.13, p <.01). Based on the rejected model, the hypothesis was not supported for the NCC. Hypothesis four stating that the PSS increases the level of intention to leave the UK has been supported. There is a positive correlation between PSS and intention to leave the UK ($\lambda$ = 44, p <.01), Hypothesis five stating that PSS decreases the level of PWB and life satisfaction has been supported. The results show a moderate, negative correlation between PSS and PWB ($\lambda$ = -.67, p <.001), PSS and life satisfaction ($\lambda$ = -.32, p <.001).

## Discussion

Our study provides the evidence-based understanding of factors affecting personal well-being of Polish immigrants living in the UK in the face of a significant political change –the Brexit vote. Although the previous study shows that there is a notably rising number of racist hate crimes committed towards immigrants after Brexit vote [3], only a small percentage of the participants reported negative changes in attitude or behaviour of supervisors (9%) and co-workers (14%) towards them. It is worth mentioning that the participants noted a positive change (7%) for their supervisors—only a two percentage points lower in comparison to a negative change. More importantly, if we take co-workers into account, a percentage for negative changes is higher (14%) and, in turn, lower for positive aspects of their behaviour and attitude towards each other. While looking for a reasonable explanation, one should note that people tend to have more co-workers than supervisors which simply means higher exposure to behavioural outcomes. From more theoretical perspective, some prevailing theories on management and leadership may provide explanation for the above-mentioned results. It has been well-established that supervisors influence subordinates' working conditions, well-being, and job strain, directly and indirectly [57, 58]. A longitudinal study conducted by Winkler, Busch,

Clasen and Vowinkel [59] confirms that changes in leadership behaviours predict changes in job satisfaction and well-being in low-skilled workers. The researchers stated that subordinates' and supervisors' individual power distance orientations will moderate the effect of subordinates' perceptions of leadership behaviour and the subsequent effects on their well-being. This is especially valid when it comes to a highly culturally diverse work setting. Building on the leader-member exchange theory which sustains that high-quality exchanges create more opportunities for interactions, work support, and the exchange of organizational and job-related information, subordinates with high power distance orientation do not undermine authority. In turn, those with low power distance orientation, expect more equality in hierarchical levels and maintain more personalized relationships with authority figures [60]. Our research finding suggests that there is a positive relation between perceived stress of employees and a negative change in their supervisors and co-workers' attitude or behaviour. This result may have a two-fold explanation. On the one hand, experiencing NCS and NCC may trigger stress reactions as, in the light of unfavourable Brexit news, an immigrant may perceive any behavioural change as a potential threat which, in turn, induces stressful reactions. NCS and NCC may be displayed through changes in fair procedures and following the relational model of procedural justice developed by Tyler and Lind [61], it may have an impact on relational bonds among people and group authorities, e.g., supervisors, managers. On the other hand, on the basis of uncertainty management theory [19], the mechanisms of fairness judgments affecting work behaviours may be explained through the concept of uncertainty. Polish migrants experiencing uncertainty due to a rapid political change may focus explicitly on fairness information which may relate to the trustworthiness of the authority in a decision-making situation. On the other hand, our research shows that only 18% variance of PSS is explained by NCC and NCS which may be interpreted that there were other factors influencing intensified perceived stress—or possibly stressful reactions may have occurred as a prime factor. Brexit itself may have induced the general feeling of being in an unfavourable position as immigrant workers which stands in line with a study of Gumbrell-McCormick and Hyman [16] where Polish workers were classified as "labour threat" to UK citizens. With this in mind, the Brexit vote may have elevated their stress level connected with unstable working conditions. As a result, their attitude, expectations, performance in workplace may have changed onto more suspicious and prone to any comments made by supervisors and co-workers. Research shows that the psychological culmination of identity and status changes for immigrants is likely to be a constant source of insecurity: perceived uncertainty about the continuation or viability of one's circumstances—inside and outside work [62]. Generally speaking, job insecurity has negative outcomes, such as increased withdrawal, declining job attitudes and performance, and poor mental and physical health [63]. The issue of authority and its perception by employees may exemplify an interesting result that social support plays a stress-reducing role for NCS but not for NCC. When a supervisor displays a negative change in his or her behaviour towards an employee, this may trigger more stressful reaction than in the case of a change of behaviour of a co-worker. Supervisors are representatives of authority of the organization. Building on social exchange theory [64], Eisenberger and colleagues [65] proposed that employees are attentive to cues about whether they are supported by their organizations. When organizations provide favourable treatment, employees feel that the organization cares about their contributions and values their well-being, which motivates them to reciprocate with strengthened affective commitment, enhanced performance, increased citizenship, and decreased withdrawal. The extensive research has corroborated these propositions by presenting that support from supervisors is a primary contributor to feeling supported by the organization [66]. Our study showed that PSS decreases the level of PWB and life satisfaction. These findings stand in line with the previous research which shows that PSS is negatively associated

with PWB, resilience, life satisfaction, and mindfulness [67, 68]. High level of stress decrease self-protection mechanisms against negative stimuli and situations. Persons who experience constant stress have a lower level of resistance, which affects other psychological resources. PSS is significantly correlated with subjects' subjective cognitive patterns [69]. The results of our study support the literature in this regard. Perceived stress of Polish migrants has been found to intensify their intention to leave the UK. It casts a new light on the recent public statistics which show that there has been a rise in the number of European citizens leaving the UK since Brexit vote. Those who experience a possible threat to their everyday existence, focus on relocation to ensure the safety of their own and their families. Referring this finding to an organisational context, [39] have found that perceived organisational support fully mediates the existing relationship between stress and intention to leave. The study conducted by Arnoux-Nicolas, Sovet, Lhotellier, Di Fabio, and Bernaud [70] shows that stressful working conditions (including, e.g. work pressure, lack of resources, organisational changes) were positively and significantly associated with turnover intentions. On the other hand, it has to be mentioned that our study explored whether the intention to leave UK has been intensified after the Brexit vote which could simply mean that the intention to move countries existed beforehand. Those who intend to leave UK may experience more stress in general which corresponds with the research stating that moving countries is a stressful life transition [71]. Job insecurity and uncertainty about the future may be linked with settling down in a new place or going back to a homeland. Summing up, Polish and British co-existence in a workplace setting has not changed much after the Brexit vote. Polish migrant workers reported almost the same percentage of negative and positive changes in attitude and behaviour of co-workers and supervisors—both were low. In turn, it may be assumed that British supervisors and co-workers did not display any form of discriminant or antisocial behaviour towards this minority group. Possibly these two cultures of workers have established a bond which stands above any politically induced divisions.

## Future research directions

Our major limitations relate to the work tenure of participants which we did not take into account and which may have affected participants' perception on work relations. Future research may explore possible negative changes in the work environment due to external political conditions through the lenses of interactional justice and supervisory ratings of antisocial work behaviours. Additionally, fairness judgments should be taken into consideration in the light of the social comparison theory. It would also be interesting to conduct a more in-depth study indicating specific antisocial behaviours experienced by immigrants. When it comes to the PWB of immigrants, it would be worth to undertake a study investigating the role of psychological resources of an individual applied, for instance, in the form of coping strategies. Future research could also benefit from exploring a mediating role of social support more in depth.

## Author Contributions

**Conceptualization:** Klaudia Martynowska.

**Data curation:** Tomasz Korulczyk.

**Formal analysis:** Piotr Janusz Mamcarz.

**Investigation:** Klaudia Martynowska.

**Methodology:** Tomasz Korulczyk.

**Supervision:** Piotr Janusz Mamcarz.

**Visualization:** Tomasz Korulczyk.

**Writing – original draft:** Klaudia Martynowska.

**Writing – review & editing:** Piotr Janusz Mamcarz.

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
