## [Decision Letter · Decision Letter 0]

29 Apr 2020

PONE-D-20-04196

Perceived stress and well-being of Polish migrants in the UK after Brexit vote

PLOS ONE

Dear Ph.D. Mamcarz,

Thank you for submitting your manuscript to PLOS ONE. After careful consideration, we feel that it has merit but does not fully meet PLOS ONE’s publication criteria as it currently stands. Therefore, we invite you to submit a revised version of the manuscript that addresses the points raised during the review process.

Your paper has been assessed by two acknowledged experts in the area covered by this study. Overall, you will find that the manuscript has received many criticism in regard to key issues such as the theoretical-methodological assumptions of the study and the interpretation of the data. Accordingly, one of the reviewers has suggested its immediate rejection, while the other asks for a major set of (extensive) revisions.

Hence, I would like to leave to the authors the possibility of revising the manuscript, in order to send it to the reviewers for their reconsideration, if you decide to submit it.

We would appreciate receiving your revised manuscript by Jun 13 2020 11:59PM. To enhance the reproducibility of your results, we recommend that if applicable you deposit your laboratory protocols in protocols.io, where a protocol can be assigned its own identifier (DOI) such that it can be cited independently in the future. For instructions see: http://journals.plos.org/plosone/s/submission-guidelines#loc-laboratory-protocols

We look forward to receiving your revised manuscript.

Kind regards,

Sergio A. Useche, Ph.D.

Academic Editor

PLOS ONE

Journal Requirements:

https://doi.org/10.1016/j.sbspro.2015.04.122

In your revision ensure you cite all your sources (including your own works), and quote or rephrase any duplicated text outside the methods section. Further consideration is dependent on these concerns being addressed.

"The funders had no role in study design, data collection and analysis, decision to

publish, or preparation of the manuscript."

Reviewers' comments:

Reviewer's Responses to Questions

**Comments to the Author**

1. Is the manuscript technically sound, and do the data support the conclusions?

Reviewer #1: Yes

Reviewer #2: No

2. Has the statistical analysis been performed appropriately and rigorously? 

Reviewer #1: I Don't Know

Reviewer #2: N/A

3. Have the authors made all data underlying the findings in their manuscript fully available?

Reviewer #1: No

Reviewer #2: No

4. Is the manuscript presented in an intelligible fashion and written in standard English?

Reviewer #1: No

Reviewer #2: No

5. Review Comments to the Author

Reviewer #1: The heart of this article is a good experiment, which is certainly worthy of publication. There are several flaws that mean that the article will require substantial revision before this should occur.

First, the standard of English is not adequate, though it is close. Most paragraphs had at least one phrase which suggested it was written by a non-native speaker and which interfered with swift interpretation. At times, I was unable to be sure I understood the authors' intended meaning. Due to the number of errors, I have not attempted to correct them, but suggest the authors engage a professional editor.

Second, the authors have assumed that the Brexit vote led to negative changes in attitudes toward Polish workers, and that this led to a range of negative psychological consequences.

I thought one of the most interesting findings was the extent to which the Brexit vote did not lead to negative changes in attitudes toward Polish workers. The proportion reporting a negative change for their supervisors was only two percentage points higher than reporting a positive change, and both were low. No statistical testing was conducted, and it isn't immediately obvious what sort would be most appropriate. Indeed, this might depend on what one intended for the results. Looking at this result as a measure of how British society has reacted to the vote suggests that it hasn't changed the perspectives of supervisors much at all. The relatively small number of respondents who reported any change in either direction is notable, and the bipolarity of this movement is notable. On the face of it, this looks like a null hypothesis to me, since one might expect that at least a small proportion of people would report a change even if there were not one. One might also expect that some people would assume the direction would be negative, and that this might account for the small difference.

The difference was more marked in the change in the attitudes of coworkers, but still not overwhelming.

On the whole, these results are worth discussing, not least in the context of the literature which argues that right-wing anti-globalisation will increase stress for minorities (see eg Macgregor-Bowles and Bowles, 2017. Trump, Brexit, Right-wing Anti-globalisation, and An Uncertain Future for Public Health). In short, the authors should integrate their findings with the social science literature beyond psychology, especially in a multi-disciplinary journal like PLOS ONE.

The difference in the apparent reactions to Brexit from co-workers and supervisors is especially interesting. It could be due to several factors. Most basically, people tend to have more co-workers than supervisors, so there are more opportunities to perceive a negative change, and this is important when most people don't perceive any change at all. Additionally, supervisors may feel less threatened by Polish workers as they do not represent such intense competition for jobs (and may represent a pool of relatively cheap, but highly skilled labour). Alternatively, supervisors may operate in social circles in which xenophobia is more sharply condemned than their workers. Another possibility is that supervisors may also be aware of their positions of authority and the need to moderate their behaviour or could more strictly constrain their behaviour. Alternatively, they may simply be more attuned to the possibility that their Polish workers may be at risk of feeling marginalised following the Brexit vote. Assessing the results by geographic area of the respondent would be interesting.

Third, in setting up and interpreting the experiment, the authors have assumed the direction of the causal arrow. They assume that Poles who say that they have experienced a negative change in the attitude or behaviour of supervisors have actually experienced this. While this is almost certainly true in some cases, it should not be assumed for all cases. It might be that people who experience elevated levels of perceived stress or decreased levels of wellbeing are more likely to interpret the words or actions of their supervisors as discriminatory. If Brexit has added to their stress due to feeling generally unwanted by the UK, or uncertainty about their futures, then it could be that they will project this general feeling onto the people in their lives such as coworkers and supervisors. It might also be that people whose stress was increased by Brexit began to perform less well at work, leading to comments or behaviours that they then interpreted as discriminatory. It is plausible that these pathways account for a reasonable amount of the observed correlations, and they need to be discussed in the paper.

The authors drew a parallel between intention to leave a work place and intention to leave the UK. While there may be some parallels, they are clearly different. There is a vast literature on immigration between countries, and this should be used to inform and support the authors' analysis.

This is another area where the causal arrow might point two directions. Perhaps those already intending to leave the UK perceive more stress. Moving countries is stressful! It is also possible that those intending to leave the UK perceive discrimination more readily as confirmation bias. This also needs discussion.

Fourth, a little more information about the methods would be useful. Specifically, how was it confirmed that Polish participants had British supervisors, and that they were talking about British coworkers (one can imagine that many had coworkers from the UK and a variety of other countries)? Additionally, how did the authors ensure that participants had been in their jobs long enough to make good observations about changes in their supervisor and coworkers?

This relates to a fifth issue, which is that the authors assume that the Brexit vote would be a clear turning point for an increase in racist behaviour. However, the lead up to the Brexit vote, with its rhetoric around immigrants, probably means that the Brexit vote itself was not a really decisive break with the preceding months. Those looking for an excuse to be racist could seek cover in the need to debate Brexit. On the whole, this might mean that the results, by relying on a comparison between the Brexit vote and the months before the Brexit vote, could underestimate the overall effect of the rise of anti-immigration sentiment on Polish workers in the UK.

Sixth, the authors could consider whether social support moderated the effect of changes in coworkers but not supervisors because coworkers are in fact a primary pool of potential social supports for immigrants. Therefore, if you coworkers are treating you less well, one of your primary potential sources of social support is no longer available.

In general, the statistical testing looked reasonable. Some of the modelling was beyond my capacity to judge, though I note that it might rely on the causal arrow pointing only one way, which seems inappropriate given what I've written above.

Reviewer #2: Thank you for your submission. The submission does not seem to have a substantive paper. Thank you for your submission. The submission does not seem to have a substantive paper. The result is not significant.

6. PLOS authors have the option to publish the peer review history of their article (what does this mean?). If published, this will include your full peer review and any attached files.

Reviewer #1: No

Reviewer #2: No

---

## [Author Response · Author response to Decision Letter 0]

9 Jun 2020

Journal Requirements:

Ad 1. (Dear Editor I have transfer the manuscript file to Letex Plos One template)

https://doi.org/10.1016/j.sbspro.2015.04.122

In your revision ensure you cite all your sources (including your own works), and quote or rephrase any duplicated text outside the methods section. Further consideration is dependent on these concerns being addressed. 

Ad 2. (Dear Editor we have addressed all the publication used in the manuscript.)

Ad 3. (Dear Editor after the consultation with our supervisors we made Our data available without request https://doi.org/10.6084/m9.figshare.12357593.v1)

Ad 3. (We will update your Data Availability statement on your behalf to reflect the information you provide.)

(We published SPSS and M+ datasets and syntaxes related with the study on Open Access Figshare repository: https://doi.org/10.6084/m9.figshare.12357593.v1)

"The funders had no role in study design, data collection and analysis, decision to

publish, or preparation of the manuscript." 

a. Please clarify the sources of funding (financial or material support) for your study. List the grants or organizations that supported your study, including funding received from your institution.

d. If you did not receive any funding for this study, please state: “The authors received no specific funding for this work.”

Ad 4. (It was misunderstanding because there was no funder involved in this study. We were participating in all the costs connected with this research.)

(Dear Editor thank You for Your effort and all the suggestion)

Reviewers' comments:

Reviewer's Responses to Questions

Comments to the Author

1. Is the manuscript technically sound, and do the data support the conclusions?

Reviewer #1: Yes

Reviewer #2: No

2. Has the statistical analysis been performed appropriately and rigorously? 

Reviewer #1: I Don't Know

Reviewer #2: N/A

3. Have the authors made all data underlying the findings in their manuscript fully available?

Ad 3. ( SPSS and M+ datasets and syntaxes related with the study are published and available here: https://doi.org/10.6084/m9.figshare.12357593.v1)

Reviewer #1: No

Reviewer #2: No

4. Is the manuscript presented in an intelligible fashion and written in standard English?

Reviewer #1: No

Reviewer #2: No 

Ad 4. (Dear Reviewers we send our manuscript to language editor. We hope that the level of English will be satisfying.)

5. Review Comments to the Author

Reviewer #1: The heart of this article is a good experiment, which is certainly worthy of publication. There are several flaws that mean that the article will require substantial revision before this should occur.

First, the standard of English is not adequate, though it is close. Most paragraphs had at least one phrase which suggested it was written by a non-native speaker and which interfered with swift interpretation. At times, I was unable to be sure I understood the authors' intended meaning. Due to the number of errors, I have not attempted to correct them, but suggest the authors engage a professional editor.

Ad 5. (Dear Reviewer we made a language editing, so we hope that the level of English will be satisfying, and the number of errors is now reduced.)

Second, the authors have assumed that the Brexit vote led to negative changes in attitudes toward Polish workers, and that this led to a range of negative psychological consequences.

Ad 5. (Dear Reviewer Thank You for pointing this out. It was not our intention to show Brexit as negative phenomenon. We changed some parts of text to add more positive context and be closer to objective perspective.)

I thought one of the most interesting findings was the extent to which the Brexit vote did not lead to negative changes in attitudes toward Polish workers. The proportion reporting a negative change for their supervisors was only two percentage points higher than reporting a positive change, and both were low. No statistical testing was conducted, and it isn't immediately obvious what sort would be most appropriate. Indeed, this might depend on what one intended for the results. Looking at this result as a measure of how British society has reacted to the vote suggests that it hasn't changed the perspectives of supervisors much at all. The relatively small number of respondents who reported any change in either direction is notable, and the bipolarity of this movement is notable. On the face of it, this looks like a null hypothesis to me, since one might expect that at least a small proportion of people would report a change even if there were not one. One might also expect that some people would assume the direction would be negative, and that this might account for the small difference.

The difference was more marked in the change in the attitudes of coworkers, but still not overwhelming.

On the whole, these results are worth discussing, not least in the context of the literature which argues that right-wing anti-globalisation will increase stress for minorities (see eg Macgregor-Bowles and Bowles, 2017. Trump, Brexit, Right-wing Anti-globalisation, and An Uncertain Future for Public Health). In short, the authors should integrate their findings with the social science literature beyond psychology, especially in a multi-disciplinary journal like PLOS ONE.

Ad 5. (Dear Reviewer. We agree with Your arguments and this can be a limitation of this research. In our revised manuscript we were trying to take a broader perspective that reflects the social nature of this phenomenon.)

The difference in the apparent reactions to Brexit from co-workers and supervisors is especially interesting. It could be due to several factors. Most basically, people tend to have more co-workers than supervisors, so there are more opportunities to perceive a negative change, and this is important when most people don't perceive any change at all. Additionally, supervisors may feel less threatened by Polish workers as they do not represent such intense competition for jobs (and may represent a pool of relatively cheap, but highly skilled labour). Alternatively, supervisors may operate in social circles in which xenophobia is more sharply condemned than their workers. Another possibility is that supervisors may also be aware of their positions of authority and the need to moderate their behaviour or could more strictly constrain their behaviour. Alternatively, they may simply be more attuned to the possibility that their Polish workers may be at risk of feeling marginalised following the Brexit vote. Assessing the results by geographic area of the respondent would be interesting. 

Ad 5. (Dear Reviewer we assessed the differences in perceived change in attitude/behaviour between region of residence of respondents. There are no significant statistical differences between regions in change in coworkers and supervisors in post hoc Befferoni tests.)

We thank the reviewer for this interesting observation and conclusion. We applied this in the discussion.

Third, in setting up and interpreting the experiment, the authors have assumed the direction of the causal arrow. They assume that Poles who say that they have experienced a negative change in the attitude or behaviour of supervisors have actually experienced this. While this is almost certainly true in some cases, it should not be assumed for all cases. It might be that people who experience elevated levels of perceived stress or decreased levels of wellbeing are more likely to interpret the words or actions of their supervisors as discriminatory. If Brexit has added to their stress due to feeling generally unwanted by the UK, or uncertainty about their futures, then it could be that they will project this general feeling onto the people in their lives such as coworkers and supervisors. 

Response: (Due to cross-sectional nature of the study, we had tested the alternative model with perceived stress and well-being being independent variables in the casual model. The model has been identified but there are no statistically significant relationships. Thus, we cannot empirically assume that people who experience elevated levels of perceived stress or decreased levels of wellbeing are more likely to interpret the words or actions of their supervisors as discriminatory and at the same time, report it in our study.)

It might also be that people whose stress was increased by Brexit began to perform less well at work, leading to comments or behaviours that they then interpreted as discriminatory. It is plausible that these pathways account for a reasonable amount of the observed correlations, and they need to be discussed in the paper.

Response: (We didn’t include such mediator as job performance in our study, thus we cannot test the hypothesis proposed by the reviewer of mediating role of job performance between perceived stress and perceived negative behaviour. However, we do agree that in some cases this relationship might be possible.) 

This is another area where the causal arrow might point two directions. Perhaps those already intending to leave the UK perceive more stress. Moving countries is stressful! It is also possible that those intending to leave the UK perceive discrimination more readily as confirmation bias. This also needs discussion.

Response: (We also estimated the model with intention to leave the UK being independent variable for negative attitude change of coworkers and supervisor with mediating role of perceived stress and well-being. The goodness-of-fit indexes are not supporting the model. Thus, the hypothesis of the reviewer that intending to leave the UK perceive more stress and then result in perceived negative attitude changes.)

Fourth, a little more information about the methods would be useful. Specifically, how was it confirmed that Polish participants had British supervisors, and that they were talking about British coworkers (one can imagine that many had coworkers from the UK and a variety of other countries)? 

Response: (We measures this using two questions indicating the possible change in behaviour or attitude of their British coworkers or supervisors:

Have you noticed a change in the attitude/behaviour of your British co-workers towards you after the Brexit referendum?

• I have seen a very positive change in the behaviour of British coworkers

• I have seen a positive change in the attitude/behaviour of British coworkers

• I have not noticed any change in the attitude/behaviour of British coworkers

• I have seen a negative change in the attitude/behaviour of British coworkers

• I have seen a very negative change in the attitude/behaviour of British coworkers

Have you noticed a change in the attitude/behaviour of your British supervisors towards you after the Brexit referendum?

• I have seen a very positive change in the attitude/behaviour of British supervisors

• I have seen a positive change in the attitude/behaviour of British supervisors

• I have not noticed any change in the attitude/behaviour of British supervisors

• I have seen a negative change in the attitude/behaviour of British supervisors

• I have seen a very negative change in the attitude/behaviour of British supervisors

Additionally, how did the authors ensure that participants had been in their jobs long enough to make good observations about changes in their supervisor and coworkers?

Response: (Unfortunately, we haven’t measured the work tenue. However, we were interested in the subjective perception of the changes of participants regardless the employment length. We agree with the reviewer it would be interesting to see if work tenue have any relationship with change in behaviour observations.)

---

## [Decision Letter · Decision Letter 1]

1 Jul 2020

Perceived stress and well-being of Polish migrants in the UK after Brexit vote

PONE-D-20-04196R1

Dear Dr. Mamcarz,

We’re pleased to inform you that your manuscript has been judged scientifically suitable for publication and will be formally accepted for publication once it meets all outstanding technical requirements.

Kind regards,

Sergio A. Useche, Ph.D.

Academic Editor

PLOS ONE

Additional Editor Comments (optional):

Reviewers' comments:

Reviewer's Responses to Questions

**Comments to the Author**

1. If the authors have adequately addressed your comments raised in a previous round of review and you feel that this manuscript is now acceptable for publication, you may indicate that here to bypass the “Comments to the Author” section, enter your conflict of interest statement in the “Confidential to Editor” section, and submit your "Accept" recommendation.

Reviewer #2: All comments have been addressed

2. Is the manuscript technically sound, and do the data support the conclusions?

Reviewer #2: Yes

3. Has the statistical analysis been performed appropriately and rigorously? 

Reviewer #2: Yes

4. Have the authors made all data underlying the findings in their manuscript fully available?

Reviewer #2: Yes

5. Is the manuscript presented in an intelligible fashion and written in standard English?

Reviewer #2: Yes

6. Review Comments to the Author

Reviewer #2: Thank you for the next opportunity to review the article "Perceived stress and well-being of Polish migrants in the UK after Brexit vote”. In my opinion, the authors have corrected the manuscript in accordance with the recommendations. So I have no more comments.

7. PLOS authors have the option to publish the peer review history of their article (what does this mean?). If published, this will include your full peer review and any attached files.

Reviewer #2: No

---

## [Editor Report · Acceptance letter]

8 Jul 2020

PONE-D-20-04196R1 

Perceived stress and well-being of Polish migrants in the UK after Brexit vote 

Dear Dr. Mamcarz:

I'm pleased to inform you that your manuscript has been deemed suitable for publication in PLOS ONE. Congratulations! Your manuscript is now with our production department. 

Kind regards, 

on behalf of

Dr. Sergio A. Useche 

Academic Editor

PLOS ONE